# Late Cambrian geomagnetic instability after the onset of inner core nucleation

Yong-Xiang Li [1] ✉, John A. Tarduno [2,3,4], Wenjun Jiao [1], Xinyu Liu[1], Shanchi Peng[5], Shihua Xu[1], Aihua Yang[1] & Zhenyu Yang[6]

The Ediacaran Period marks a pivotal time in geodynamo evolution when the geomagnetic field is thought to approach the weak state where kinetic energy exceeds magnetic energy, as manifested by an extremely high frequency of polarity reversals, high secular variation, and an ultralow dipole field strength. However, how the geodynamo transitioned from this state into one with more stable field behavior is unknown. Here, we address this issue through a high-resolution magnetostratigraphic investigation of the ~494.5 million-year-old Jiangshanian Global Standard Stratotype and Point (GSSP) section in South China. Our paleomagnetic results document zones with rapid reversals, stable polarity and a ~80 thousand-year-long interval without a geocentric axial dipole field. From these changes, we suggest that for most of the Cambrian, the solid inner core had not yet grown to a size sufficiently large to stabilize the geodynamo. This unusual field behavior can explain paleomagnetic data used to define paradoxical true polar wander, supporting instead the rotational stability of the solid Earth during the great radiation of life in the Cambrian.

Earth's inner core nucleation (ICN) represents a major change in core structure and in how the geodynamo is powered[1,2]. The ICN age was once thought to be many billions of years old[3,4], but more recently, studies of the thermal evolution of Earth, core thermal conductivity and paleomagnetism raise the possibility of a relatively young age, less than 1 billion years old[1,5–9]. In particular, the thermal evolution and geodynamo model of Driscoll[1] predicts that the geodynamo reached the weak-field state, when core kinetic energy exceeds magnetic energy, prior to and during ICN. The weak-field state is unstable, characterized by an extremely high frequency of polarity reversals (also called the hyper-reversal state), high directional dispersion and very low field intensity. Independently, several workers had documented evidence for a hyper-reversal state in the Ediacaran Period[10–12]. The reversal rate is estimated to be >20 reversals per Myr in the late Ediacaran[12,13], which is 3 to 5 times more frequent than the reversal frequency of, on average, ~4–5 reversals per Myr during Mesozoic and younger times. But these studies did not directly bear on the most critical aspect preceding ICN in the model predictions of Driscoll[1]: dipole field strength. A separate independent effort on field strength during the Ediacaran Period culminated in a report of an ultralow field intensity for 565 Ma rocks[7], some ten times less than the present-day field strength. The ultralow field is the weakest time-averaged geomagnetic field strength known over the entire history of the geodynamo[14–17]. Because of this extremely low value and coeval unusual directional behavior, Bono et al.[7] linked the Ediacaran to ICN. This ultralow field has been supported by multiple subsequent studies where instantaneous field recordings of the Ediacaran field by dikes and lavas[18,19] have been reported. Recently, Zhou et al.[9] report a time-averaged field 5 times stronger than the Ediacaran ultralow value for the early Cambrian (532 Ma). This further supports the Ediacaran as the time of ICN because the field is expected to rapidly strengthen as new energy sources become available to power the geodynamo[9]. Other

[1]State Key Laboratory for Mineral Deposits Research, Institute of Continental Geodynamics, School of Earth Sciences and Engineering, Nanjing University, Nanjing 210023, China. [2]Department of Earth & Environmental Sciences, University of Rochester, Rochester, NY, USA. [3]Department of Physics & Astronomy, University of Rochester, Rochester, NY, USA. [4]Laboratory for Laser Energetics, University of Rochester, Rochester, NY, USA. [5]State Key Laboratory of Geology and Palaeontology, Nanjing Institute of Palaeontology, Chinese Academy of Sciences, Nanjing 210008, China. [6]College of Resources, Environment & Tourism, Capital Normal University, Beijing 100048, China. ✉e-mail: yxli@nju.edu.cn

times have also been proposed for ICN, but none of the data-based interpretations are grounded on time-averaged paleointensity values showing a characteristic change from the weak-field state to stronger fields (Supplementary Information, SI).

While observations continue to support the extraordinarily unstable state of the geodynamo during the Ediacaran, what is largely unknown is how the geodynamo transitioned out of this unusual state and into the more stable directional state of longer polarity chrons that marks, for example, the Mesozoic and younger times. This transition is particularly important for understanding events near the Precambrian-Cambrian boundary, because this time marks the rise of animal groups[20]. This biological evolution could be influenced by the nature of plate tectonics, but the Cambrian apparent polar wander paths have long been controversial[10,21–23]. These data have been interpreted as marking the rapid rotation of the entire solid Earth relative to the spin axis, or true polar wander (TPW), and that this motion, through associated environmental change, was a driving force for animal evolution[21]. However, it is unclear whether the dramatic climate shifts attendant with rapid TPW would facilitate or thwart evolution, and the rates of TPW needed to explain paleomagnetic observations appear to be geophysically implausible even if the thermal history of mantle is taken into account[24,25]. While TPW interpretations continue to be pursued[26] based on the observation of disparate magnetic directions, recently refined chronologies of paleopoles used to define the Ediacaran TPWs do not seem to support prior TPW interpretations[27].

Mapping reversal frequency patterns of the Ediacaran-Cambrian interval provides one means of gaining insight into the stabilization of the geodynamo following the Ediacaran ultralow intensity interval. A hyper-reversal mode proposed for the Ediacaran is thought to have persisted into the Cambrian based on the magnetostratigraphic studies of the late Ediacaran to Cambrian strata[12,28,29]. Although absolute time control provided from radiometric age data is lacking in many investigations, the occurrence of many reversals in several localities with biostratigraphic age constraints lends support for an unusually high reversal rate. These localities include the Zigan Formation in southern Urals, Russia[12,13], the Lopata Formation in southwest Siberia[30], the Lena River section of Siberia[28,31], the Precambrian-Cambrian sequences in Chengjiang area in eastern Yunnan, China[32,33], and the early Cambrian Niutitang Formation in South China[34]. An estimate from the Zigan Formation with a refined chronology using cyclostratigraphy suggests ~20 reversals per Myr[13]. A high frequency of reversals is also documented in the Middle Cambrian sequences in Siberia[29,35–37] with an estimate of 10 reversals per Myr from the Kulyumbe River section (NW Siberia)[37] and up to 26 reversals per Myr from the Khorbusuonka section (NE Siberia)[29]. By the Cambrian-Ordovician boundary, reversal rates are estimated to decrease to ~1 reversal per Myr[37]. Therefore, the late Cambrian appears to have recorded substantial changes in the nature of the geodynamo.

Well-dated, continuous late Cambrian successions are perhaps best preserved in South China where two global stratotype sections and points (GSSPs) within the late Cambrian Furongian Series are established, namely the GSSPs for Jiangshanian Stage and the Paibian Stage[38]. The GSSP sections in South China thus present a unique opportunity to investigate the reversal frequency patterns in the late Cambrian. A recent paleomagnetic study across the Jiangshanian GSSP reveals the dominance of a normal polarity in Jiangshanian Stage and only a single site (Site DB13) of reversed polarity below the base of the Jiangshanian GSSP in the Paibian Stage[39].

Herein, we carry out a high-resolution magnetostratigraphic investigation of the Jiangshanian GSSP section with a particular focus on the site of reversed polarity and nearby sediments in the succession[40]. The ~26 m thick succession studied straddles the Jiangshanian GSSP in South China, with the GSSP being at 19.6 m. The section belongs to the Huayansi Formation that consists of limestone and shaly limestone deposited on the outermost part of the northeast-

striking Jiangnan slope (Supplementary Fig. S1; SI). Sample spacing is balanced by the need to maintain the Geo-heritage status of the section and samples from 195 stratigraphic levels were analyzed ("Methods").

## Results and discussion

### Magnetostratigraphy: high-frequency polarity reversals, stable polarity, and unusual field behavior

Thermal demagnetization reveals that the vast majority of samples exhibit a three-component magnetization[41] (Fig. 1). The low-, intermediate-, and high-temperature component (LTC [<~200 °C], ITC [~200 °C to ~350 °C], and HTC [~350 °C to ~500 °C]) structure is similar to that reported in the previous study[39]: the LTC documents a recent overprint and the ITC represents a Mesozoic overprint. The HTC, or the characteristic remanent magnetization (ChRM), exhibits dual polarities (Fig. 1) (Supplementary Table S1). Rock magnetic data including thermal demagnetization of 3-component composite isothermal remanent magnetization (IRM) and magnetic hysteresis loops suggest that magnetite grains are the major remanence carriers and the dominant magnetic grains are in pseudosingle domain (PSD) state or a mixture of single domain (SD) and multi-domain (MD) states (Supplementary Figs. S3, S4). Scanning electron microscopy with electron dispersive spectroscopy ("Methods", SI) documents the presence of detrital magnetite grains, many of which show small Al or Cr peaks (Supplementary Fig. S5 Part 2). Al and Cr substitution lowers the Curie temperature of magnetite, consistent with the dominant unblocking at 500 °C relative to that of end-member magnetite

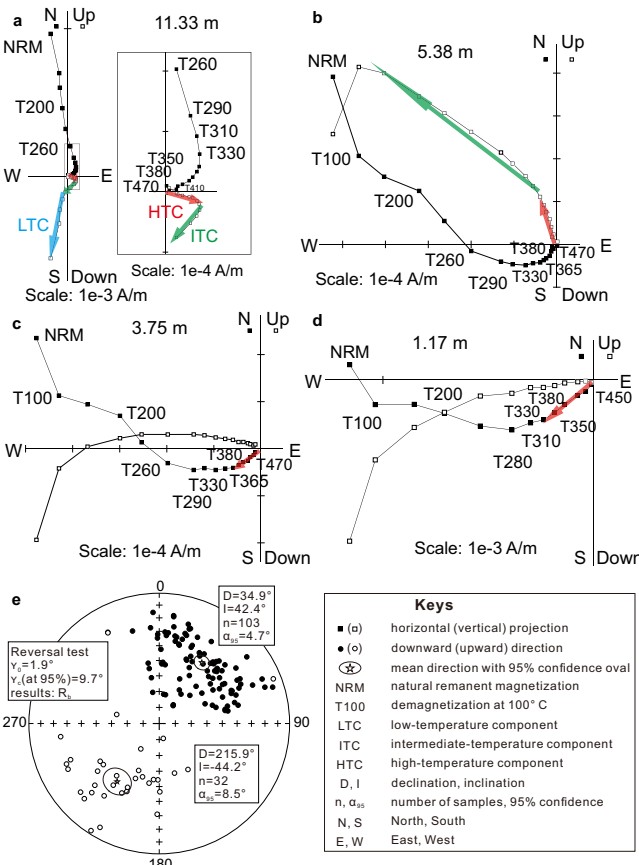

**Fig. 1 | Representative demagnetization plots in geographic coordinates.** **a–d** Stepwise thermal demagnetization data showing three-component magnetizations. **e** Reversal test of the high-temperature components (HTCs). The antipodal directions pass the reversals test at 95% confidence level with "B" class[45].

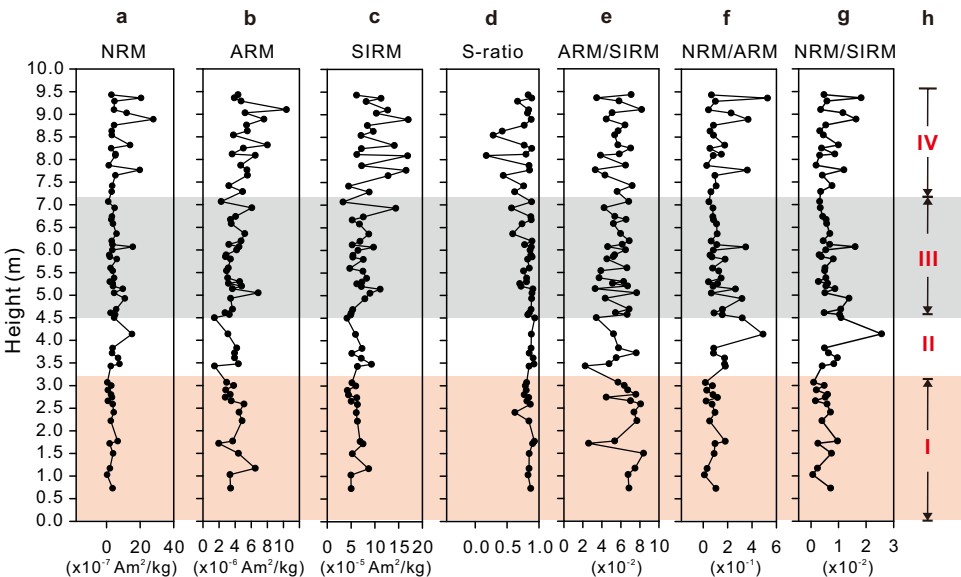

**Fig. 2 | Detailed rock magnetic stratigraphy (0–9.5 m). a–e** Changes in NRM, natural remanent magnetization (**a**), ARM, anhysteretic remanent magnetization (**b**), SIRM, saturation isothermal remanent magnetization (**c**), S-ratio (**d**), and ARM/SIRM (**e**) with stratigraphic height. S-ratio is defined as IRM acquired at 300 mT backfield over SIRM acquired at 1.0 T forward field. **f, g** NRM normalized by ARM and SIRM, respectively, as a crude relative paleointensity proxy. **h** Four intervals (I, II, III, IV) with different geomagnetic field behavior character documented in the studied section (see Fig. 3).

(-580 °C). In addition, evidence for detrital titanomagnetite is also observed (Supplementary Fig. S5 Part 5c).

A series of rock magnetic parameters ("Methods") of the 0 to ~9.5 m interval (Fig. 2) (Supplementary Table S2) show no dramatic changes that could result from secondary magnetic minerals. A positive fold test for the paleomagnetic data from both the NW and SE limbs of the fold in this area (Supplementary Fig. S1) suggests that the studied interval preserves a record of the Cambrian geomagnetic field[39].

The paleomagnetic data of the studied interval show a sequence of magnetic polarity changes with stratigraphic height. Below ~3.5 m, the paleomagnetic data show either northerly downward directions, indicative of normal polarities, or southerly upward directions, indicative of reversed polarities. These data indicate four pairs of normal and reversed polarity zones below ~3.5 m (Fig. 3), suggesting frequent polarity reversals during this interval. The stratigraphic levels between ~3.5 and ~4.5 m show persistently southerly declination and upward inclinations, defining a reversed polarity zone (Fig. 3). From ~4.5 to ~7.1 m the magnetic directions are predominantly upward, but the associated virtual geomagnetic poles (VGPs) cannot be assigned to reversed or normal polarity. As we will describe below, these define a very different type of geomagnetic behavior than previously reported in the literature. Above ~7.1 m, the northerly declinations and downward inclinations are persistently present, defining a normal polarity zone.

To assess the geomagnetic significance of these data, we first obtain the average sedimentation rate of ~3.2 cm/kyr for the Paibian Stage using the thickness of the Paibian Stage in the section[42] and its upper and lower bound ages from the Cambrian Period of the Geological Time Scales[38,43,44] (SI). We then use the estimated sedimentation rate to assign durations to the 4 intervals with different field behavior character. From the base of the sampled sequence to the top, these intervals and durations are as follows (Fig. 3f, g): (I) An interval of hyper reversals spanning ~109 kyrs between 0 and ~3.5 m. (II) An interval of stable reversed polarity between ~3.5 m and ~4.5 m spanning ~31 kyrs; (III) An interval between ~4.5 m and 7.1 m of highly unusual field behavior spanning ~80 kyrs; and (IV) A stable normal polarity interval between ~7.1 and ~26 m that spans at least 590 kyrs.

For Interval I, we estimate the reversal frequency to be ~36 reversals per Myr. A rapid reversal frequency has been reported from lower Cambrian drill cores in South China[34] and recently demonstrated also in an outcrop section in Serbia[29]. A middle Cambrian rapid reversal rate has also been reported from these studies, mainly in the Drumian Stage[29]. The hyper-reversal feature in our magnetostratigrahic sequence occurs in the Paibian Stage in the late Cambrian. Thus, it appears that intervals of rapid or hyper-reversals occurred throughout the Cambrian, and that Interval I represents the youngest hyper-reversal feature of the Cambrian. Intervals II and IV of our section appear to represent more stable periods of reversed and normal polarity of the Cambrian geodynamo, respectively. A McFadden and McElhinny's reversal test performed on the ChRMs from these intervals passes at 95% confidence level with class "B"[45] (Fig. 1e), further supporting the stability of the field. Interval II marks a major reversed polarity below the Cambrian Jiangshanian GSSP that was defined by the first appearance (FA) of the cosmopolitan agnostoid trilobite *Agnostotes orientalis*[40]. Based on the mean sedimentation rate of the Paibian Stage (SI), Interval II occurred at ~470 kyr before the start of the Jiangshanian Stage. Since the Jiangshanian GSSP has also been shown to occur near the termination of the Steptoean positive carbon isotope excursion (SPICE)[46], Interval II, together with the GSSP trilobite zones and the carbon isotope chemostratigraphy, provides powerful additional constraints for refining the late Cambrian geological timescale and establishing global correlations of the chronostratigraphy around the Jiangshanian Stage.

### A longstanding non-geocentric axial dipole field

Interval III shows unparalleled geomagnetic behavior. We note that the lithology and sedimentary features (e.g., composition, texture, structures, etc.) within Interval III are the same as those bounding sediments, indicating that abnormal sedimentary processes (e.g., slumping, turbidity current, etc.) are not a factor. In addition, rock magnetic data such as ARM, SIRM, S-ratio, etc. do not show anomalous changes in magnetic mineralogy, concentration, and grain size that could otherwise indicate any compromised recording ability of magnetic mineral carriers in Interval III (Fig. 2a–e). And SEM and EDS

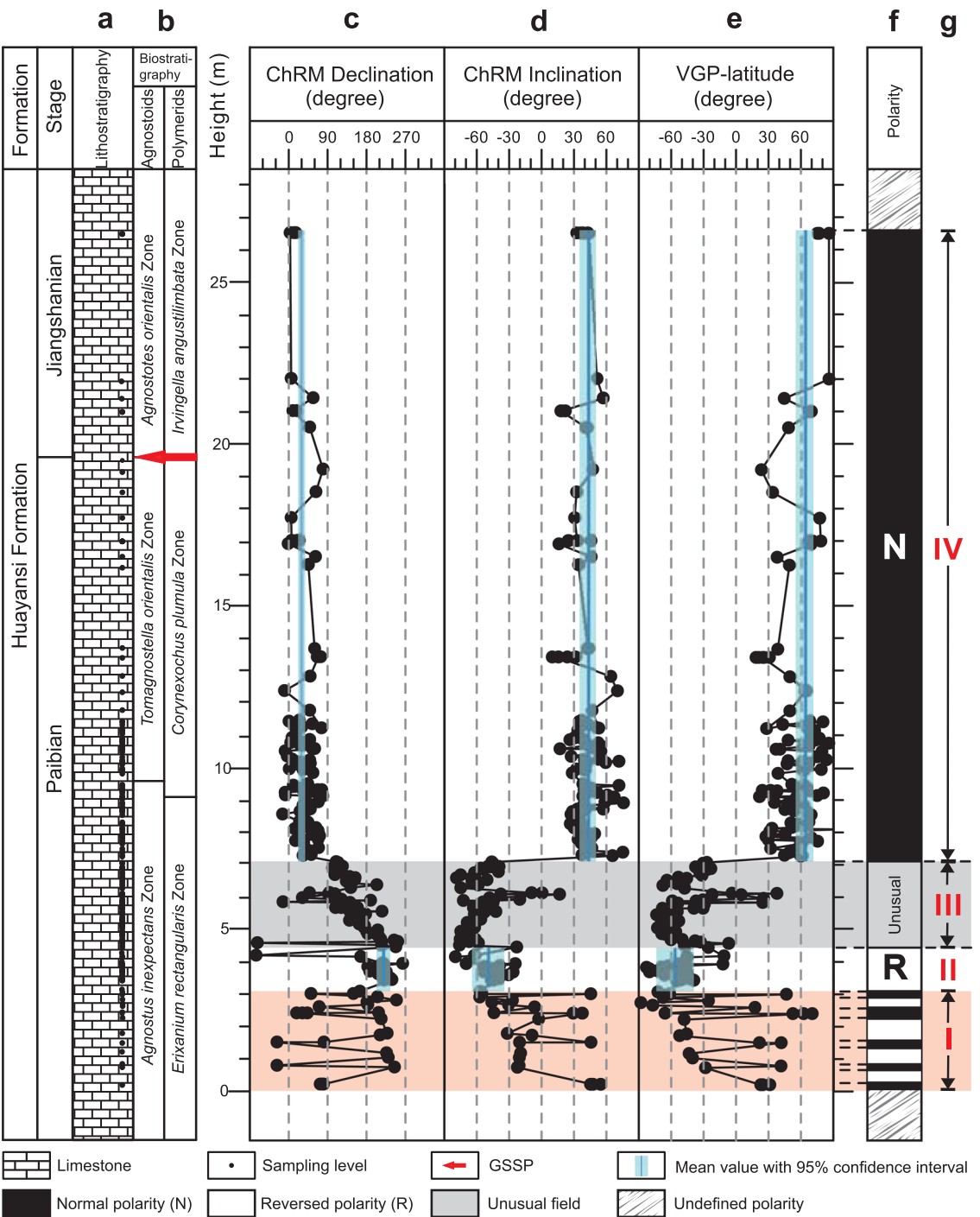

**Fig. 3 | The integrated litho-, bio-, and magnetostratigraphy of the late Cambrian Jiangshanian GSSP section. a** Lithostratigraphy[40] shows that the studied section consists of limestone. **b** Biostratigraphy[40] indicates the Jiangshanian GSSP. **c**–**e** Changes in the declination and inclination of characteristic remanent magnetization (ChRM), and the virtual geomagnetic pole (VGP)-latitude with stratigraphic levels, respectively. **f** Established magnetostratigraphy for the GSSP section; N, normal polarity; R, reversed polarity. **g** Four intervals (I, II, III, IV) with different geomagnetic field behavior character documented in the studied section. The stratigraphic height follows that marked at the Jiangshan GSSP Geopark.

properties of the sediments are also similar to those of the bounding intervals, except for a few outliers noted below.

The directional behavior of Interval III is distinct from transitional behavior between polarity states observed in younger sediments in two key ways. First, Cenozoic polarity transitions have been estimated to last between ~2 and 22 kyrs[47,48]; Interval III is at least 3.6 times longer. Second, Cenozoic polarity reversals are characterized by substantial decreases in total field strength. While absolute paleointensity data and precise relative paleointensity (RPI) data are not available, the normalized NRM data (Fig. 2f, g) as a crude RPI proxy, where NRM is by and large proportional to $NRM_{350\,°C}$ around which the start of ChRM is chosen (Supplementary Fig. S6; SI), do not detect any anomalously decreased field intensity in Interval III.

To further investigate the field behavior of Interval III, we applied cluster analysis[49] to the VGPs to assess the average field behavior of Interval III, which yielded 3 dominant modes (Fig. 4). Interestingly, in stratigraphic height, the modes define an arc of nearly 90° from the stable reversed polarity (Interval II) mean VGP, to the point of furthest

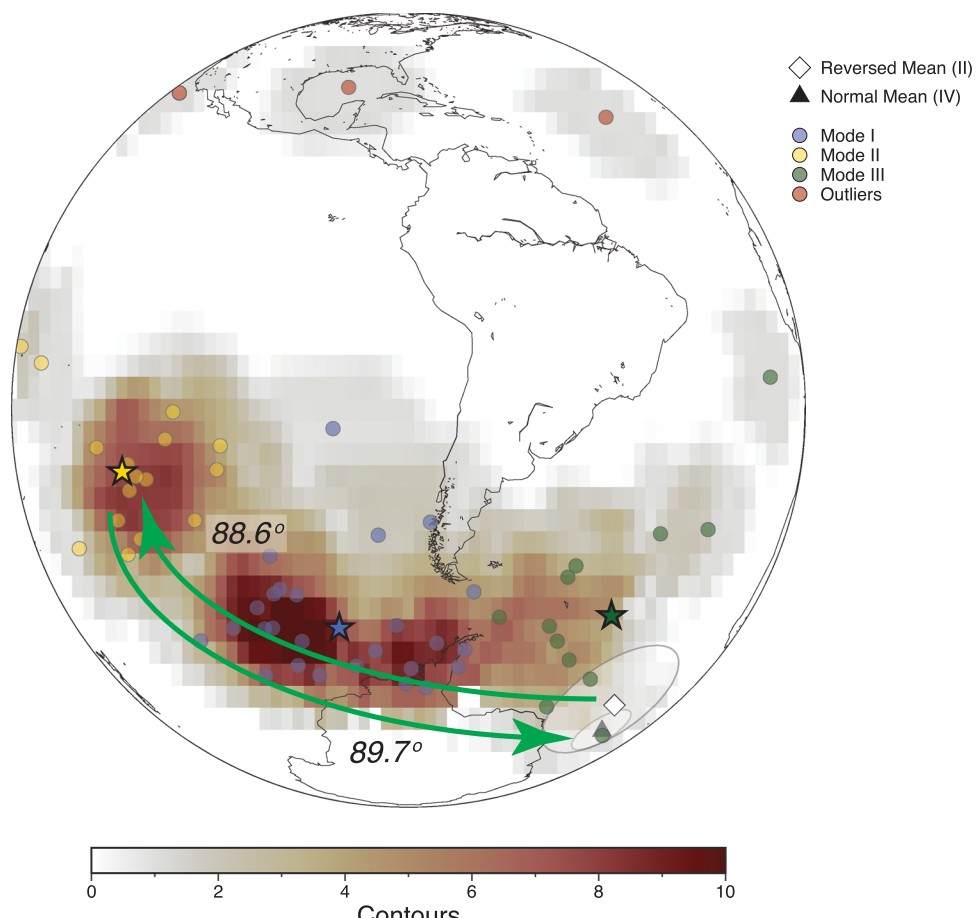

**Fig. 4 | Non-geocentric axial dipole field (Interval III).** Spherical density distribution and k-means cluster analysis of virtual geomagnetic poles (VGPs) following Bono et al.[49], shown in orthographic projection centered on −30°, 280° E. Spherical contouring uses a uniform grid of nodes with a median distance of 4° and a counting radius of 10°. Color intensity corresponds with the density of directions within the counting radius (value shown in color scale). Circles: individual VGPs, color corresponds with cluster mode assignment. Red circles show directions identified as outliers. Stars: k-means cluster centers for each mode. Modes are as follows: Mode I (blue) 257.9° E, −62.3°; Mode II (gold/yellow) 224.9° E, −28.2°; Mode III (green) 337.1° E, −52.4°. Open diamond: Stable reversed polarity mean VGP from the Jiangshanian GSSP section (Interval II) with $A_{95}$ circle of confidence. Solid triangle: Stable normal polarity VGP mean antipode (Interval IV), with $A_{95}$ circle of confidence. Green arrows highlight the angular distance between the stable reversed polarity VGP and Mode II, and the angular distance between Mode II and the stable normal polarity (antipode) VGP.

angular departure (mode II), where the pole resides for >10 kyr. The pole then moves rapidly back, without intermediate modes, almost exactly 90° to the stable normal polarity (antipode) mean pole (Interval IV). The few outliers observed from this pattern (Fig. 4) appear to have anomalous magnetic carriers (SI). Although a global view of magnetic directions is needed to constrain field morphology, the mode II field is consistent with a long-lasting equatorial dipole field[50]. Modes I and III also represent longstanding non-axial dipole states, ~20° and 49°, respectively, from the stable reversed polarity mean pole.

## An unstable dynamo and its implications from core to the surface

The character of the field, with switches between stable (Intervals II, IV), unstable (i.e., Interval I, hyper-reversal), and the longstanding non-axial dipole behavior (Interval III), may be a characteristic of the core during the first ~70 Myr of the inner core growth, and ultimately related to the geometry of outer core flow. For example, in numerical simulations, Lhuillier et al.[51] found polar downwellings toward the inner core when the inner core was small, and upwellings from the inner core for larger inner cores. Using thermal evolution models[6] and a Late Ediacaran growth onset age[7], we predict the inner core radius to be ~35 to 40% of its current value during the time interval represented

by our data. On the basis of the changes in character of the geomagnetic field we observe, we suggest that the inner core had not grown large enough in the early to late Cambrian to produce a core flow pattern that would stabilize the geodynamo, but that this stabilization likely occurred in the latest Cambrian (Paibian/Jiangshanian stages) to early Ordovician times. We note that after this stabilization, the Earth saw increases in marine biodiversity and ocean-atmospheric oxygen levels, and the emergence of geomagnetic field states characteristic of the Phanerozoic (Supplementary Fig. S9). Irrespective of the increasing evidence supporting an Ediacaran ICN age, it is important to emphasize that our hypothesis linking core size and field stability generally applies for all plausible ICN onset ages equal to or older than the Ediacaran Period (SI). That is, an inner core older than our preferred Ediacaran age would expand the time range containing shorter intervals of field instability from that hypothetical older ICN initiation age to the late Cambrian.

The geocentric axial dipole (GAD) hypothesis[52,53] is fundamental to using paleomagnetic data for reconstructing plate configurations, plate velocities and true polar wander, all of which have been proposed as playing key environmental controls ultimately driving the Cambrian diversification of life. Our data indicate that at times in the Cambrian the GAD hypothesis does not hold. Our results indicate that individual paleomagnetic results of Cambrian age, especially those from

relatively rapidly cooled igneous rocks like lavas or dikes[26], must be interpreted with caution because rather than recording a GAD field, they could record a highly non-axial geocentric dipole field coherent on timescales of ten thousand years or longer (Fig. 4)[10,54] and paleolatitude estimates from the paleomagnetic data during a non-GAD period could be incorrect[55].

These findings have a special meaning for the proposal of Cambrian inertial interchange TPW (IITPW)[21], whereby the entire Earth rotated by 90° causing environmental change, but at rates incompatible with mantle viscosity constraints[24,25]. Specifically, a sampling of an Interval III mode II field 90° away from the stable normal and reversed polarity states could be misinterpreted to represent IITPW. Sampling of fields similar to the other modes of Interval III could be mistaken to represent smaller amounts of TPW[56] (Fig. 4). Thus, an unstable dynamo producing a non-axial geocentric dipole field when the inner core was small, can resolve the geophysical paradox posed by the posit of IITPW. Therefore, our observations of field instability are consistent with a rotationally stable Cambrian Earth[10]. This stability, without huge, rapid shifts of latitudinal belts, would have allowed life to thrive and evolve gradually[57], allowing the great diversification of life on the planet.

## Methods

### Sampling

To better define the polarity zonation, paleomagnetic samples were collected every ~5–10 cm near and within the reversed polarity site reported from the previous study of the Jiangshanian GSSP section[39] (Supplementary Fig. S1). Away from this site, sample spacing was gradually increased to ~0.5 or 1.0 m. One core was typically collected from one bed although sometimes two cores were taken from one horizon. We note that samples from different lateral positions within the same bed likely sample slightly different vertical heights (see Paleomagnetic results in SI). Paleomagnetic samples were collected with a portable rock drill and were oriented using a magnetic compass mounted into a Pomery orientation device. Also, ten oriented hand samples were collected for compliance with regulations prohibiting drilling of the outcrop after the GSSP Geopark was established. Samples from 195 stratigraphic levels were collected from the ~26 m succession. In addition, bedding attitudes were measured for tilt-corrections of the samples in the subsequent paleomagnetic data analysis.

### Paleomagnetic measurements and analyses

In the laboratory, the oriented block samples were drilled and all the core samples were trimmed to standard (2.54 cm diameter × 2.2 cm height) cylindrical paleomagnetic specimens. Demagnetization experiments with sister specimens at ~19 m of the section using both alternating field (AF) and thermal technique indicate that thermal demagnetization is more effective than AF demagnetization (Supplementary Fig. S2). Therefore, all the specimens were then subjected to stepwise thermal demagnetization using an ASC TD48 thermal demagnetizer. The remanence of specimens was measured with a three-axis, 2G Enterprise Inc. 755 DC-SQUID rock magnetometer housed in a magnetically shielded room (residual field <300 nT) in the Paleomagnetism Laboratory of Nanjing University, China. The demagnetization data are plotted with vector-end-point diagrams[58] and principal component analysis (PCA)[59] was performed to define magnetization components. Components with a maximum angular deviation (MAD) < 15° are considered acceptable. Software packages Puffinplot[60,61] and PMGSC (by Randy Enkin) and PaleoMac[62] were utilized for paleomagnetic data analysis. Figure 4 was made using PyGMT (Generic Mapping Tools)[63], an open-source collection of command-line tools for manipulating geographic and Cartesian data sets; coastline and other geospatial data sets are freely available.

### Rock magnetic experiments

To constrain magnetic mineralogy of remanence carriers, several types of rock magnetic measurements were conducted for selected specimens. Composite isothermal remanent magnetization (IRM) was acquired subsequently along Z-, Y-, and X-axis at 1.2 T, 0.5 T, and 0.125 T, respectively, using an ASC IM−30 impulse magnetizer. The composite IRM was then subjected to progressive thermal demagnetization and the remanence was measured with a JR-6A spinner magnetometer[64]. In addition, residuals from trimming the standard cylindrical paleomagnetic specimens from the 0–9.5 m interval were used to impart a series of magnetization to further examine magnetic mineralogy characteristics of this interval. An anhysteretic remanent magnetization (ARM) was acquired in an alternating field (AF) peaked at 100 mT and a bias field of 0.05 mT using a Molspin demagnetizer coupled with pARM apparatus. A saturation isothermal remanent magnetization (SIRM) was imparted with an ASC IM−30 impulse magnetizer at 1.0 T and the SIRM is subsequently subjected to a backfield applied at 300 mT, acquiring an isothermal remanent magnetization at −300 mT, i.e., IRM-300. The IRMs were measured with an AGICO JR-6A spinner magnetometer. The NRM normalized by ARM or SIRM provides a rough gauge of relative paleointensity to detect whether there is any anomalous intensity in the studied interval. The ARM/SIRM ratio provides a measure of abundance of fine-grained magnetic particles capable of carrying stable remanence. We also calculate the S-ratio, which is defined as the ratio of IRM−300/SIRM. The S-ratio is particularly sensitive to magnetic mineralogy as expressed as changes in magnetic coercivity. Magnetic hysteresis loops were measured with a MicroMag 3900 alternating gradient magnetometer in Institute of Geology and Geophysics, Chinese Academy of Science (CAS).

### Petrographic analysis

Petrographic observations of the samples were carried out with the Zeiss Supra 55 field emission scanning electron microscope (FE-SEM) coupled with an Oxford Aztec X-Max 150 energy dispersive spectrometer (EDS) at Nanjing University, China. Back-scattered SEM images and EDS point analyses and elemental mapping were used to examine magnetic minerals. The instrument was operated at a 15-kV accelerating voltage during the SEM observations and EDS analyses. The samples were coated with either Au or Pt.

## Data availability

The paleomagnetic data generated in this study has been deposited in the Earthref (MagIC) database (earthref.org/MagIC/19613; https://doi.org/10.7288/V4/MAGIC/19613). Summary of the paleomagnetic data and rock magnetic data generated in this study are provided in the Supplementary Information (Supplementary Tables S1, S2).

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

## Acknowledgements

We thank Shipeng Wang, Lifeng Ma, Siyi Xie, and Boxing Zhang for field assistance. Qiwei Qian helped measure some pilot samples. We thank Richard Bono and Rory Cottrell for cluster analysis. This study was supported by the National Natural Science Foundation of China (41230208 Z.Y., 41888101 Y.-X.L., 41774075 Y.-X.L.). J.T. acknowledges support from the US National Science Foundation (1828817). We are grateful to the Jiangshan Geopark authority for allowing access to the outcrop.

## Author contributions

Conceptualization: Y.-X.L. and J.T.; investigation: Y.-X.L., W.J., S.P., X.L., S.X., and A.Y.; formal analysis: Y.-X.L., W.J., J.T., X.L., S.X., S.P., and Z.Y.; funding acquisition: Z.Y. and Y.-X.L.; writing—original draft: Y.-X.L. and J.T.; writing—reviewing and editing: Y.-X.L., J.T., W.J., S.P., A.Y., and Z.Y.

## Competing interests
