## [Peer Review File · Nature Communications]

Late Cambrian geomagnetic instability after the onset of inner core nucleationREVIEWER COMMENTS

Reviewer #1 (Remarks to the Author):

Comments on the manuscript "Late Cambrian geomagnetic instability and its bearing on Earth's rotational stability" by Yong-Xiang Li et al.

In this paper, Yong-Xiang Li and colleagues deal with a very fascinating discussion related to the development of the solid inner core and the consequent stabilization of the geodynamo.

They report a high-resolution magnetostratigraphic survey of the Jiangshanian Global Standard Stratotype and Point (GSSP) section of about 494.5 million years ago in southern China. Their results document four distinct intervals characterized by different field behaviour character: (1) an area with rapid reversals, (2) an area with stable polarity, (3) an area without a geocentric axial dipole field and (4) an interval with stable normal polarity.

The authors attribute the observed changes to a progressive accretion, during the Cambrian, of the solid inner core to a size large enough to stabilize the geodynamo. In addition, the interval (3) characterized by about 80 kyrs without a geocentric axial dipole field poses an important criticism to the reconstruction of the True Polar Wander (the solid-body rotation of the entire Earth with respect to its spin axis) at times during the Cambrian.

The authors have done a very good job in many respects of this paper. I like it. In my view, the interpretation is a plausible and good explanation of the observation. The paper is certainly of interest and my feeling is that it meets the criteria for acceptance in Nature Comm.

I suggest minor changes before it is considered for publication.

- Figure 3c and 3d : at the top of these two figures, I suggest to write "ChRM Declination" and "ChRM Inclination"

- Figure 3 I see that the sampling resolution within Zone I is different from Zone II and III. How come? Is this perhaps due to an inappropriate lithology for the portable rock drill? I am asking this because this is the range with many inversions and some of these are defined only on a single sample... Personally I am skeptical when I see single samples with different inclination from the surrounding samples but this is a particular context and so I would just like to understand better.

- Figure 5a: NRM is in A/m but in Figure 2 it is in A/m/kg

- about the units for magnetization: why did you use a volume magnetization normalized by mass (A/m/kg) and not just the volume magnetization (A/m) or the mass magnetization (Am²/kg)?

- line 152-153 : I READ "Two samples from 8.7 m and 7.87 m show hysteresis loops with a saturation field of ~300 or 400 mT (Extended Data Fig. 4), suggesting the presence of magnetic minerals of high coercivity, which again may point to a contribution of hematite." FROM THE PLOT I DO NOT SEE THIS

- line 166-167: you write "Second, ARM and IRM, the normalizers, of the rock magnetic samples exhibit strong correlation" CAN YOU PROVIDE A R² ?

- Line 201: DO YOU MEAN "... in the older part of the Cambrian"?

- line 203-204: "Limestone shows relatively weak magnetism and thermal alteration could occur during lab heating that would cause remanence increase" HOW DO YOU INTERPRET THIS INCREASE IN REMANENCE?

Finally, I see that for paleomagnetic data analysis you used Puffinplot. Please note that the most updated version is the following:

Lurcock, P.C., Florindo, F., 2019. New developments in the Puffin Plot paleomagnetic data analysis program. *Geochem. Geophys. Geosyst.* 20, 5578–5587. <https://doi.org/10.1029/2019GC008537>

Reviewer #2 (Remarks to the Author):

General Comment

This paper presents an analysis of new sedimentary data from China in the early Cambrian. They identify periods of anomalous geomagnetic instability and frequent reversals, consistent with observations from earlier Precambrian-Cambrian boundary and Ediacaran. The authors hypothesize that the core was in an anomalous state at this time because the inner core was still small, which agrees with some models.

My only general comment is regarding the connection the authors draw between the paleomagnetic data and the previously published ideas of true polar wander during this time period. The authors make a valid point that paleogeographic models should acknowledge the possibility of true dipole wander (that is non-GAD behavior) and should encourage more investigation into the paleomagnetic signature of a non-GAD field vs TPW. The title of the paper draws undue attention to this aspect with the phrase "and its bearing on Earth's rotational stability". This rather vague phrase is somewhat misleading. In my opinion the paper would be better received by focusing on its own results: that the early to mid Ordovician geomagnetic field shows directional anomalies similar to those found in the Cambrian and late Neoproterozoic. This alone is an important discovery. The connection to TPW is more of a speculation at this point, albeit an important one.

Detailed Comments

Lines 124-126: What about the data implies magnetite? What about the data implies PSD or SD/MD?

Lines 170-171: What "lithology and sedimentary features" are you referring to? What features would indicate an abnormal sedimentary process if one occurred?

Lines 172-173: What about the magnetic data indicates that the recording ability of the magnetic carriers does not change? Please be more specific.

Lines 193-194: "... may be a characteristic of the core during its first ~70 Myr of growth..." By "core" do you mean "inner core"?

Lines 205-207: "it is important to emphasize that our hypothesis linking core size and field stability generally applies for all plausible ICN onset ages equal to or older than the Ediacaran Period." The data presented here is from the Cambrian, so how does it relate to the geomagnetic field of the Ediacaran or older? Please clarify.

Lines 214-215: If the field was non-GAD, what would the magnetic carriers record? This question has been addressed (in part) by Olson et al. (2018) and Driscoll & Wilson (2018) (ref's below).

Lines 222-223: "Therefore, our observations of field instability are consistent with a rotationally stable Cambrian Earth." This seems like an overreach at this point. Sure, non-GAD field should be acknowledged, if not accounted for, in TPW models. But how do you do that? A more careful investigation into the paleomagnetic signature of non-GAD vs TPW seems necessary to even address this question.

Peter Olson, Maylis Landeau, Evan Reynolds, True dipole wander, *Geophysical Journal International*, Volume 215, Issue 3, December 2018, Pages 1523–1529, <https://doi.org/10.1093/gji/ggy349>

P. E. Driscoll and C. Wilson. Paleomagnetic biases inferred from numerical dynamos and the search for geodynamo evolution. *Frontiers in Earth Science*, 6:113, 2018. <https://doi.org/10.3389/feart.2018.00113>

Reviewer #3 (Remarks to the Author):

The inner core nucleation and true polar wander are two hotly debated topics in paleomagnetic studies. Recent paleomagnetic studies consistently indicate inner core nucleation in the Ediacaran Period characterized by extremely high frequency of polarity reversals, high secular variation, and ultralow dipole field strength. How the geodynamo transitioned from this state prior to and during inner core nucleation into a stable geocentric axial dipole state is fascinating. Another intriguing and controversial question of the Ediacaran and Cambrian periods is the observed polar wander, which is usually interpreted to indicate rapid rotation of the entire solid Earth relative to the spin axis. This, if true, could have induced dramatic environmental and climate changes affecting animal evolution. The reversal frequency of the geomagnetic field determined from magnetostratigraphic studies of well-dated and continuously deposited sedimentary strata is a valuable parameter to estimate the stability of the geodynamo. In this research, Li et al. present a high-resolution magnetostratigraphic investigation of the Jiangshanian GSSP section in South China, which provide the opportunity to decipher such puzzles.

Compared to other magnetostratigraphic studies on this topic, the age of the Jiangshanian GSSP section is well constrained to ~494.5 Ma, extending to the late Cambrian. The reported paleomagnetic and rock magnetic data reported in this research are of high quality, the analyses of these data are robust, and the results are well presented. The authors identify hyper reversals interval of 109 kyrs, stable reversed polarity interval of ~31 kyrs, a ~80 thousand-year-long interval without a geocentric axial dipole field, and a following stable normal polarity interval of >590 kyrs upsection. These findings suggest that the inner core has not yet grown to a size sufficiently large to stabilize the geodynamo in Cambrian and that such unusual field behavior can explain paleomagnetic data used to define paradoxical true polar wander. This work will be a great contribution to the paleomagnetic study of the inner core evolution and true polar wander, and will also interest a broad community of geoscientists working on geodynamo, paleontology, plate tectonics, and environmental changes. I fully agree with the publication of this work in Nature Communications.

My major suggestion to the manuscript is limited to strengthen the rock magnetic part and provide SEM/TEM observations to firmly support the primary origin of the HTC isolated from these carbonates. The preservation of a primary remanence magnetization in these carbonates is the base of this research. Although I agree that positive fold and reversals tests presented in this research strongly suggest a primary origin of the HTC, a complete and convincing demonstration of this requires robust rock magnetic and petrographic evidences. Actually, the wasp-waisted hysteresis loops and distribution of the ratios of hysteresis parameters in or near the remagnetization region of carbonates in the Day plot show risk of remagnetization of these rocks, which reminds that the carrier(s) and acquisition processes of remanence (LTC, ITC, HTC) within these rocks should be carefully discussed. To argue that remagnetization is restricted to the LTC and ITC and that the HTC is of primary origin, additional experiments might be required. I would suggest the authors to conduct low-temperature susceptibility or magnetization experiments to see if Verwey transition can be identified. Verwey transition at 120 K is usually smeared in remagnetized carbonate, which is dominated by oxidized authigenic magnetite (Jackson and Swanson-Hysell, 2012). If the magnetic carrier of the HTC is detrital (titano)magnetite or biogenic magnetite, which can carry a primary remanence, Verwey transition should be detected. Furthermore, SEM or TEM observations can verify the existence of detrital or biogenetic magnetite and authigenic magnetic minerals in these rocks. I also notice that the authors applied TEM observations on magnetic extracts from carbonates rocks of the lower and middle Cambrian strata in this region (Jiao et al., 2018), but this has not been applied on the upper Cambrian strata. SEM/TEM observations, when coupled with complete rock magnetic experiments, will greatly help to understand the carriers and origins of remanence in carbonates from the upper Cambrian strata shown in this manuscript.

Below are some minor suggestions.

Line 121: The temperature range of the LTC, ITC and HTC should be defined here.

Line 123: The authors argued for a Triassic remagnetization for the ITC carried by pyrrhotite in Jiao et al. (2018). However, this point has not been discussed in this manuscript. Is there anything

different in rock magnetism for rocks studied here and Jiao et al. (2018)? A low-temperature magnetization experiment may also help to identify pyrrhotite with Besnus transition at 32 K in these rocks. Actually, the y axis magnetization with applied field of 0.5 T in Extended Data Fig. 3a show the sign of pyrrhotite. This should be carefully discussed in the manuscript.

Line 179: The NRM is dominated by secondary remagnetization signals, I wonder if it is meaningful to use it for such an argument. Maybe the intensity of the HTC is more suitable. If there are other secondary magnetic carriers, then the stratigraphic changes of values of ARM, SIRM and their ratios in Fig. 2 should be reevaluated.

Line 184: Should inclination shallowing be considered here?

Extended data Fig. S: It seems that only 8.70 m specimen contains quite some hematite, other specimens have little concentration of hematite as indicated by saturation of the remanence before 0.5 T. For that reason, 8.70 m specimen is not suitable to plot in the Day plot because Day plot is for (titano)magnetite, and that wasp-waisted hysteresis loops of most samples do not indicate mixture of magnetite with hematite. Instead, these loops are very similar to the shape of remagnetized carbonate rocks (Jackson and Swanson-Hysell, 2012). This problem has to be addressed.

References:

- Jackson, M., Swanson-Hysell, N.L., 2012. Rock magnetism of remagnetized carbonate rocks: another look. *Geological Society, London, Special Publications* 371, 229-251.
- Jiao, W.-J., Li, Y.-X., Yang, Z.-Y., 2018. Paleomagnetism of a well-dated marine succession in South China: A possible Late Cambrian true polar wander (TPW). *Physics of the Earth and Planetary Interiors* 277, 38-54.

RESPONSE TO REVIEWER COMMENTS

Reviewer 1

Comment 1: In this paper, Yong-Xiang Li and colleagues deal with a very fascinating discussion related to the development of the solid inner core and the consequent stabilization of the geodynamo.

They report a high-resolution magnetostratigraphic survey of the Jiangshanian Global Standard Stratotype and Point (GSSP) section of about 494.5 million years ago in southern China. Their results document four distinct intervals characterized by different field behaviour character: (1) an area with rapid reversals, (2) an area with stable polarity, (3) an area without a geocentric axial dipole field and (4) an interval with stable normal polarity.

The authors attribute the observed changes to a progressive accretion, during the Cambrian, of the solid inner core to a size large enough to stabilize the geodynamo. In addition, the interval (3) characterized by about 80 kyrs without a geocentric axial dipole field poses an important criticism to the reconstruction of the True Polar Wander (the solid-body rotation of the entire Earth with respect to its spin axis) at times during the Cambrian.

The authors have done a very good job in many respects of this paper. I like it. In my view, the interpretation is a plausible and good explanation of the observation. The paper is certainly of interest and my feeling is that it meets the criteria for acceptance in Nature Comm.

Response 1: We thank the reviewer for this assessment.

Comment 2: Figure 3c and 3d: at the top of these two figures, I suggest to write "ChRM Declination" and "ChRM Inclination"

Response 2: Done.

Comment 3: Figure 3 I see that the sampling resolution within Zone I is different from Zone II and III. How come? Is this perhaps due to an inappropriate lithology for the portable rock drill? I am asking this because this is the range with many inversions and some of these are defined only on a single sample... Personally I am skeptical when I see single samples with different inclination from the surrounding samples but this is a particular context and so I would just like to understand better.

Response 3: The sampling density reflects practical constraints. Dense sampling – by drilling – is no longer possible because of the Geo-heritage status of the site, something that occurred during the different phases of the sampling described. We have reemphasized in the revised text that sample density is balanced by the need to maintain the heritage of the site. The sample spacing in Intervals II and III is mostly less than 10 cm. Given the drill core samples have a diameter of 2.54 cm, the sampling resolution is uncommonly high. The reviewer is correct in noting there are a few polarities defined by a single sample in Interval I. However, there are more polarities defined by more than one samples, and sample spacing in some cases is only a few cm, almost close to or smaller than the size of a standard cylindrical paleomagnetic specimen. Thus, when two samples were drilled from the same bed in the field, or from a block sample from that bed, the two samples from the same nominal height could in reality have cm-scale differences in height. For example, in Interval I at 3.0 m where one sample was drilled from an oriented block sample and the other was a minicore drilled in the field, one sample shows normal polarity and the other shows reversed polarity (table S1). Thus, the data most likely reflect the magnetic field at slightly different times and support a hyper-reversal state.

Comment 4: Figure 5a: NRM is in A/m but in Figure 2 it is in A/m/kg

Response 4: For Figure 5a, the reviewer probably means Figure 1a because there are only four figures in the main text. Fig. 1 shows the thermal demagnetization data of standard cylindrical paleomagnetic specimens (2.54 cm diameter, 2.2 cm height). Because the specimens that were thermally demagnetized cannot be used for rock magnetic measurements, the rock magnetic data shown in Fig. 2 were obtained by measuring the residuals when cutting a drill core into standard cylindrical paleomagnetic specimens. Therefore, rock magnetic parameters are normalized by the mass of the residual samples. As noted in the reply below to Comment 5, the unit in Fig. 2 should be Am²/kg and has been corrected in the revision.

Comment 5: about the units for magnetization: why did you use a volume magnetization normalized by mass (A/m/kg) and not just the volume magnetization (A/m) or the mass magnetization (Am²/kg)?

Response 5: Thanks for pointing this out! The unit in Fig. 2 should be Am²/kg and has been corrected in the revision.

Comment 6: line 152-153: I READ "Two samples from 8.7 m and 7.87 m show hysteresis loops with a saturation field of ~300 or 400 mT (Extended Data Fig. 4), suggesting the presence of magnetic minerals of high coercivity, which again may point to a contribution of hematite." FROM THE PLOT I DO NOT SEE THIS

Response 6: It is correct that saturation did not occur exactly at ~300 or 400 mT, but greater than ~300 or 400 mT, suggesting the presence of magnetic minerals of high coercivity. In the revision, "a saturation field of ~300 or 400 mT" has been changed to "a saturation field of > ~300 or 400 mT".

Comment 7: line 166-167: you write "Second, ARM and IRM, the normalizers, of the rock magnetic samples exhibit strong correlation" CAN YOU PROVIDE A R² ?

Response 7: r²=0.52 and is shown in Fig. 6c in the revision.

Comment 8: Line 201: DO YOU MEAN "... in the older part of the Cambrian"?

Response 8: No, the stabilization should be later. We have changed this to "latest Cambrian" to clarify.

Comment 9: line 203-204: "Limestone shows relatively weak magnetism and thermal alteration could occur during lab heating that would cause remanence increase" HOW DO YOU INTERPRET THIS INCREASE IN REMANENCE?

Response 9: It is a common observation that when carbonates and other sedimentary rocks alter during thermal demagnetization, remanence increases. However, this is a common empirical observation, and the actual processes are not well-studied. They are probably complex, involving the creation of new minerals, interaction fields and the propensity of these new minerals to acquire even trace fields in the laboratory. We have modified the sentence to read: "...remanence increase, which presumably reflects the propensity of newly formed magnetic minerals to record stray internal and external fields, ..."

Comment 10: Finally, I see that for paleomagnetic data analysis you used Puffinplot. Please note that the most updated version is the following:
Lurcock, P.C., Florindo, F., 2019. New developments in the Puffin Plot paleomagnetic data analysis program. *Geochem. Geophys. Geosyst.* 20, 5578–5587. <https://doi.org/10.1029/2019GC008537>

Response 10: Added. We thank the reviewer for noting this.

Reviewer 2

Comment 1: This paper presents an analysis of new sedimentary data from China in the early Cambrian. They identify periods of anomalous geomagnetic instability and frequent reversals, consistent with observations from earlier Precambrian-Cambrian boundary and Ediacaran. The authors hypothesize that the core was in an anomalous state at this time because the inner core was still small, which agrees with some models.

My only general comment is regarding the connection the authors draw between the paleomagnetic data and the previously published ideas of true polar wander during this time period. The authors make a valid point that paleogeographic models should acknowledge the possibility of true dipole wander (that is non-GAD behavior) and should encourage more investigation into the paleomagnetic signature of a non-GAD field vs TPW.

Response 1: Thank you very much for your time and efforts in reviewing our manuscript. We thank the reviewer for this assessment.

Comment 2: The title of the paper draws undue attention to this aspect with the phrase “and its bearing on Earth’s rotational stability”. This rather vague phrase is somewhat misleading. In my opinion the paper would be better received by focusing on its own results: that the early to mid Ordovician geomagnetic field shows directional anomalies similar to those found in the Cambrian and late Neoproterozoic. This alone is an important discovery. The connection to TPW is more of a speculation at this point, albeit an important one.

Response 2: We agree with the reviewer that the field behavior is the prominent discovery. To fully reflect this, we have changed our original title to the following:

New title: Late Cambrian geomagnetic instability after the onset of inner core nucleation

However, we note that our data are important with respect to TPW because of time scales. The duration of the field variations we observe is far too short to record TPW, but far too long to record normal paleosecular variations. However, if dikes or flows cooled and recorded this field behavior, and a long time scale was assigned (commonly done in studies of Cambrian and older lavas and dikes lacking high resolution dates), it would be very difficult or impossible to separate this from TPW. Also, the magnitude of the observed directional difference (~90°) of the non-GAD field is almost identical to the magnitude of the ITPW proposed for the early Cambrian that is within the time interval of field instability from the onset of inner core growth to late Cambrian. As such, our prominent discovery has direct bearing on TPW. Therefore, we respectfully disagree that this is speculation- it follows directly from defining field behavior and a consideration of the geological timescales of recording by dikes or basalts often used to define a TPW, particularly early Cambrian TPW.

Comment 3: Lines 124-126: What about the data implies magnetite? What about the data implies PSD or SD/MD?

Response 3: The thermal demagnetization data of 3-component composite IRMs define a magnetite, or near end-member magnetite composition, and hysteresis loops data indicate the domain state of PSD or SD/MD. We have clarified this by specifically adding the following: “Rock magnetic data including thermal demagnetization of 3-component composite isothermal remanent magnetization (IRM) and magnetic hysteresis loops suggest that...” in the text.

Comment 4: Lines 170-171: What “lithology and sedimentary features” are you referring to? What features would indicate an abnormal sedimentary process if one occurred?

Response 4: Lithology and sedimentary features including lithological composition, textures, structures etc. that could signal any disturbance to normal depositional processes. Disruption to

normal depositional processes would be indicated by a large range of factors including an influx of siliciclastic rocks, turbidites, slumps, etc. that would distort paleomagnetic records. All of these are not observed, which also makes the studied section qualified as a Geoheritage site important for the Cambrian geologic timescale (GSSP-Jiangshan stage). In the revision, we have added a few examples of the types of abnormal depositional processes that are not present in the studied section.

Comment 5: Lines 172-173: What about the magnetic data indicates that the recording ability of the magnetic carriers does not change? Please be more specific.

Response 5: In the revision, we have added a note on the specific rock magnetic parameters such as ARM, SIRM, S-ratios etc. that record this.

Comment 6: Lines 193-194: "... may be a characteristic of the core during its first ~70 Myr of growth..." By "core" do you mean "inner core"?

Response 6: Yes. Thank you for catching this! The confusion of the "core" vs "inner core" was caused by "its" and we have clarified this by changing "its first ~70 Myr of growth" to "the first ~70 Myr of the inner core growth".

Comment 7: Lines 205-207: "it is important to emphasize that our hypothesis linking core size and field stability generally applies for all plausible ICN onset ages equal to or older than the Ediacaran Period." The data presented here is from the Cambrian, so how does it relate to the geomagnetic field of the Ediacaran or older? Please clarify.

Response 7: We have clarified this statement by adding a new sentence: "That is, an inner core older than our preferred Ediacaran age would expand the time range containing shorter intervals of field instability from that hypothetical older ICN initiation age to the late Cambrian."

Comment 8: Lines 214-215: If the field was non-GAD, what would the magnetic carriers record? This question has been addressed (in part) by Olson et al. (2018) and Driscoll & Wilson (2018) (ref's below).

Response 8: This is an interesting question, but without global data we can only speculate. Specifically, any one horizon could record the global field with varying amounts of non-GAD components, but it could be even further anomalous based on the nature of the core. The reviewer may be implying that inferring site paleolatitudes would be incorrect, and that is a safe conclusion. We have added that to the manuscript and also the Olson et al. (2018) and Driscoll & Wilson (2018) references.

Comment 9: Lines 222-223: "Therefore, our observations of field instability are consistent with a rotationally stable Cambrian Earth." This seems like an overreach at this point. Sure, non-GAD field should be acknowledged, if not accounted for, in TPW models. But how do you do that? A more careful investigation into the paleomagnetic signature of non-GAD vs TPW seems necessary to even address this question.

Response 9: Earth today, and over the well-defined intervals of the recent past (say, since the Pliocene) is not undergoing large TPW. Hence the null hypothesis is that Earth has not undergone large TPW. The reviewer appears to present large TPW as the null hypothesis. In our view, large TPW is the alternative hypothesis. A compelling data is needed to exclude the null hypothesis. Our data outline a fundamental limitation of data used by others who have suggested large terrestrial TPW.

Reviewer 3

Comment 1: The inner core nucleation and true polar wander are two hotly debated topics in paleomagnetic studies. Recent paleomagnetic studies consistently indicate inner core nucleation in the Ediacaran Period characterized by extremely high frequency of polarity reversals, high secular variation, and ultralow dipole field strength. How the geodynamo transitioned from this state prior to and during inner core nucleation into a stable geocentric axial dipole state is fascinating. Another intriguing and controversial question of the Ediacaran and Cambrian periods is the observed polar wander, which is usually interpreted to indicate rapid rotation of the entire solid Earth relative to the spin axis. This, if true, could have induced dramatic environmental and climate changes affecting animal evolution. The reversal frequency of the geomagnetic field determined from magnetostratigraphic studies of well-dated and continuously deposited sedimentary strata is a valuable parameter to estimate the stability of the geodynamo. In this research, Li et al. present a high-resolution magnetostratigraphic investigation of the Jiangshanian GSSP section in South China, which provide the opportunity to decipher such puzzles.

Compared to other magnetostratigraphic studies on this topic, the age of the Jiangshanian GSSP section is well constrained to ~494.5 Ma, extending to the late Cambrian. The reported paleomagnetic and rock magnetic data reported in this research are of high quality, the analyses of these data are robust, and the results are well presented. The authors identify hyper reversals interval of 109 kyrs, stable reversed polarity interval of ~31 kyrs, a ~80 thousand-year-long interval without a geocentric axial dipole field, and a following stable normal polarity interval of >590 kyrs upsection. These findings suggest that the inner core has not yet grown to a size sufficiently large to stabilize the geodynamo in Cambrian and that such unusual field behavior can explain paleomagnetic data used to define paradoxical true polar wander. This work will be a great contribution to the paleomagnetic study of the inner core evolution and true polar wander, and will also interest a broad community of geoscientists working on geodynamo, paleontology, plate tectonics, and environmental changes. I fully agree with the publication of this work in Nature Communications.

Response 1: We thank the reviewer for this assessment.

Comment 2: My major suggestion to the manuscript is limited to strengthen the rock magnetic part and provide SEM/TEM observations to firmly support the primary origin of the HTC isolated from these carbonates. The preservation of a primary remanence magnetization in these carbonates is the base of this research. Although I agree that positive fold and reversals tests presented in this research strongly suggest a primary origin of the HTC, a complete and convincing demonstration of this requires robust rock magnetic and petrographic evidences. Actually, the wasp-waisted hysteresis loops and distribution of the ratios of hysteresis parameters in or near the remagnetization region of carbonates in the Day plot show risk of remagnetization of these rocks, which reminds that the carrier(s) and acquisition processes of remanence (LTC, ITC, HTC) within these rocks should be carefully discussed. To argue that remagnetization is restricted to the LTC and ITC and that the HTC is of primary origin, additional experiments might be required. I would suggest the authors to conduct low-temperature susceptibility or magnetization experiments to see if Verwey transition can be identified. Verwey transition at 120 K is usually smeared in remagnetized carbonate, which is dominated by oxidized authigenic magnetite (Jackson and Swanson-Hysell, 2012). If the magnetic carrier of the HTC is detrital (titano)magnetite or biogenic magnetite, which can carry a primary remanence, Verwey transition should be detected.

Response 2: We agree that the wasp-waisted curves, indicative of SP grains mixed with larger grains, move close to but not within the so called remagnetization field of carbonates. We have emphasized that SP ferric oxides from weathering can induce similar behavior, as has been documented in carbonates from the Umbrian Apennines of Italy and accreted limestones in California. We have added a citation to the work on the Umbrian red/pink limestones (Channell and McCabe, 1994). Also, low temperature tests in the context of our work are ambiguous.

These carbonates are expected to have detrital iron-bearing minerals and any substitution into the lattice will smear the Verwey transition and thus limit its utility. Indeed, as Jackson and Swanson-Hysell (2012) also pointed out, impurity can completely suppress the Verwey transition of magnetite. To test this notion, we also performed low-temperature (LT) experiment (ZFC and FC measurements) for the sample at 5.78 m where SEM data clearly show the presence of detrital magnetite grains (Extended Data Fig. 5 Part 1c), but with evidence also of Al substitution. Both the FC and ZFC curves show no Verwey transition at ~120 K (the figure below), a diagnostic feature for pure magnetite (without substitution). Because of the evidence of diverse substitution into the lattice of these detrital grains, we feel low temperature measurements are of lesser utility.

Channell, J. E. T. & McCabe, C., Comparison of magnetic hysteresis parameters of unremagnetized and remagnetized limestones *J. Geophys. Res.* **99**, 4613-4623 (1994).

Comment 3: Furthermore, SEM or TEM observations can verify the existence of detrital or biogenetic magnetite and authigenic magnetic minerals in these rocks. I also notice that the authors applied TEM observations on magnetic extracts from carbonates rocks of the lower and middle Cambrian strata in this region (Jiao et al., 2018), but this has not been applied on the upper Cambrian strata. SEM/TEM observations, when coupled with complete rock magnetic experiments, will greatly help to understand the carriers and origins of remanence in carbonates from the upper Cambrian strata shown in this manuscript.

Response 3: TEM images from the lower and middle Cambrian are shown in Fig. 7 of Jiao et al. (2018). TEM observations on magnetic extracts from the upper Cambrian were indeed conducted previously as well and are shown in Fig. 8 of Jiao et al. (2018).

However, the reviewer's comments have motivated us to conduct additional SEM analyses. We have added a new section to the supplementary information (Section 4.2) describing these new analyses and the new SEM data are shown in Extended Data Fig. 5. These new data allow us to reemphasize our prior interpretations, but also to note a few new observations in the main text. Specifically:

- 1) We confirm the presence of detrital magnetite grains in all intervals sampled.

- 2) We detect and document data suggesting Al and/or Cr substitution in the magnetite, potentially lowering Curie temperatures, and compatible with the major unblocking temperature of 500 °C.
- 3) Similarly, we find evidence for Ti, and hence the possibility of titanomagnetite with unblocking less than 580 °C.
- 4) We find evidence for rare replacement of pyrite by magnetite in a sample with an outlier direction, potentially explaining the few outlier directions we identified in Figure 4.

We feel these observations substantially strengthen the manuscript and we thank the reviewer for suggesting the additional analyses that led to them.

Comment 4: Line 121: The temperature range of the LTC, ITC and HTC should be defined here.

Response 4: Added.

Comment 5: Line 123: The authors argued for a Triassic remagnetization for the ITC carried by pyrrhotite in Jiao et al. (2018). However, this point has not been discussed in this manuscript. Is there anything different in rock magnetism for rocks studied here and Jiao et al. (2018)? A low-temperature magnetization experiment may also help to identify pyrrhotite with Besnus transition at 32 K in these rocks. Actually, the y axis magnetization with applied field of 0.5 T in Extended Data Fig. 3a show the sign of pyrrhotite. This should be carefully discussed in the manuscript.

Response 5: The thermal demagnetization data of 3-component IRM of Jiao et al (2018) are from samples collected from sites/sections different from those reported here. However, our thermal demag data of the 3-component IRM are similar to those of Jiao et al. (2018). In all of these plots, the decay of the 0.5 T component between 300-350°C is rather gradual, and not typical of the sharp change typically seen in pyrrhotite (e.g., Dunlop and Özdemir, 1997). Given the new data, we now feel the identification of pyrrhotite is uncertain, and we now note this in the revised supplement. While low temperature data might resolve the presence of some pyrrhotite, the gradual slope of the 0.5 T component of the 3-component IRM demagnetization data suggests that any overprints in this range likely also be carried by another mineral (e.g., relatively larger magnetite with Al or Cr substitution, or titanomagnetite).

Dunlop, D. J. & Özdemir, Ö. Rock Magnetism, Fundamentals and Frontiers (Cambridge University Press, New York, NY, 1997).

Comment 6: Line 179: The NRM is dominated by secondary remagnetization signals, I wonder if it is meaningful to use it for such an argument. Maybe the intensity of the HTCs is more suitable. If there are other secondary magnetic carriers, then the stratigraphic changes of values of ARM, SIRM and their ratios in Fig. 2 should be reevaluated.

Response 6: As we indicate in the main text, the normalized NRM data is considered as a crude RPI proxy to detect if there are any anomalous rock magnetic properties across interval III. Details of the argument are provided in the 2nd paragraph of section 4.1 of the supplementary information (SI). Because $NRM_{350^{\circ}C}$, around which the start of ChRM is chosen, is found to be largely proportional to NRM of the paleomagnetic specimens (Extended Data Fig. 6a) and thermally demagnetized paleomagnetic specimens cannot be used for rock magnetic measurements, rock magnetic measurements were made on residual samples and it is reasonable to normalize the NRM of residual samples to obtain a crude RPI proxy.

Besides the detailed explanations in the SI, we add the following in the main text: "...as a crude RPI proxy, where NRM is by and large proportional to $NRM_{350^{\circ}C}$ around which the start of ChRM is chosen (Extended Data Fig. 6; SI), ..."

Comment 7: Line 184: Should inclination shallowing be considered here?

Response 7: The lithology of Intervals I to III do not suggest any major changes that might lead to differential compaction and different inclination shallowing. It is true that even with a constant flattening factor inclinations could be affected differentially due to the functional relationship of inclination shallowing (e.g., Tarduno, 1990). However, the dynamic range of the VGP shift supersedes any detailed application of a correction for inclination shallowing, and clearly inclination shallowing cannot cause reversed directions and other large changes in declination or the range of inclinations observed. This has been added to the SI.

Tarduno, J.A. Absolute inclination values from deep sea sediments: a reexamination of the Cretaceous Pacific record. *Geophys. Res. Lett.* **17**, 101-104 (1990).

Comment 8: Extended data Fig. S4: It seems that only 8.70 m specimen contains quite some hematite, other specimens have little concentration of hematite as indicated by saturation of the remanence before 0.5 T. For that reason, 8.70 m specimen is not suitable to plot in the Day plot because Day plot is for (titanio)magnetite, and that wasp-waisted hysteresis loops of most samples do not indicate mixture of magnetite with hematite. Instead, these loops are very similar to the shape of remagnetized carbonate rocks (Jackson and Swanson-Hysell, 2012). This problem has to be addressed.

Response 8: As noted above, there are multiple pathways to produce wasp-waisted curves in rocks without remagnetizations. Plus positive fold and reversal tests strongly suggest a primary origin of the HTC, arguing against remagnetization. Furthermore, the hysteresis loops of Jackson and Swanson-Hysell (2012) show saturation around 0.1 T, while our hysteresis loops data show saturation around 0.3-0.4 T. This difference suggests that the wasp-waistedness of our hysteresis loops probably result from grains with contrast particle sizes. We have reemphasized this in the text. Also, following the reviewer's suggestion, we have removed the datapoint of 8.7 m sample from the Day plot.

REVIEWERS' COMMENTS

Reviewer #3 (Remarks to the Author):

It is a pleasure for me to review the revised manuscript of this research. I really appreciate the authors' accomplishment of the low-temperature magnetic measurements, SEM observations, and EDS analyses that I suggested. The additional work further confirms the primary origin of the HTC in most rocks. It also provides a reasonable explanation for the abnormal magnetic behavior of the few outliers. Their interpretation of the absence of Verwey transition in the new acquired low-temperature data is also convincing and supported by the EDS results. Besides the critical finding of the Late Cambrian geomagnetic instability, this research also provides a successful case of isolating and verifying the primary remanence in carbonate rocks, which is usually challenging. The paleomagnetic community will benefit from the research methods used by the authors. For these reasons and other great improvement of the manuscript by the authors, I highly recommend the editor to accept the manuscript for publication in its present revised form.